# Multiple Mechanisms Confer Resistance to Azithromycin in *Shigella* in Bangladesh: a Comprehensive Whole Genome-Based Approach

Suraia Nusrin,[a] Asaduzzaman Asad,[b] Shoma Hayat,[b] Bitali Islam,[a] Ruma Begum,[b] Fahmida Habib Nabila,[b] Zhahirul Islam[b]

[a]Department of Genetic Engineering and Biotechnology, East West University, Dhaka, Bangladesh
[b]Laboratory of Gut-Brain Signaling, Laboratory Sciences and Services Division, icddr,b, Dhaka, Bangladesh

Suraia Nusrin and Asaduzzaman Asad contributed equally to this paper as first authors, and their author order was determined by drawing straws.

**ABSTRACT** *Shigella* is the second leading cause of diarrheal deaths worldwide. Azithromycin (AZM) is a potential treatment option for *Shigella* infection; however, the recent emergence of AZM resistance in *Shigella* threatens the current treatment strategy. Therefore, we conducted a comprehensive whole genome-based approach to identify the mechanism(s) of AZM resistance in *Shigella*. We performed antimicrobial susceptibility tests, polymerase chain reaction (PCR), Sanger (amplicon) sequencing, and whole genome-based bioinformatics approaches to conduct the study. Fifty-seven (38%) of the *Shigella* isolates examined were AZM resistant; *Shigella sonnei* exhibited the highest rate of resistance against AZM (80%). PCR amplification for 15 macrolide resistance genes (MRGs) followed by whole-genome analysis of 13 representative *Shigella* isolates identified two AZM-modifying genes, *mph*(A) (in all *Shigella* isolates resistant to AZM) and *mph*(E) (in 2 AZM-resistant *Shigella* isolates), as well as one 23S rRNA-methylating gene, *erm*(B) (41% of AZM-resistant *Shigella* isolates) and one efflux pump mediator gene, *msr*(E) [in the same two *Shigella* isolates that harbored the *mph*(E) gene]. This is the first report of *msr*(E) and *mph*(E) genes in *Shigella*. Moreover, we found that an IncFII-type plasmid predominates and can possess all four MRGs. We also detected two plasmid-borne resistance gene clusters: IS*26-mph*(A)-*mrx*(A)-*mph*(R)(A)-IS*6100*, which is linked to global dissemination of MRGs, and *mph*(E)-*msr*(E)-IS*482*-IS*6*, which is reported for the first time in *Shigella*. In conclusion, this study demonstrates that MRGs in association with pathogenic IS*6* family insertion sequences generate resistance gene clusters that propagate through horizontal gene transfer (HGT) in *Shigella*.

**IMPORTANCE** *Shigella* can frequently transform into a superbug due to uncontrolled and rogue administration of antibiotics and the emergence of HGT of antimicrobial resistance factors. The advent of AZM resistance in *Shigella* has become a serious concern in the treatment of shigellosis. However, there is an obvious scarcity of clinical data and research on genetic mechanisms that induce AZM resistance in *Shigella*, particularly in low- and middle-income countries. Therefore, this study is an approach to raise the alarm for the next lifeline. We show that two key MRGs [*mph*(A) and *erm*(B)] and the newly identified MRGs [*mph*(E) and *msr*(E)], with their origination in plasmid-borne pathogenic islands, are fundamental mechanisms of AZM resistance in *Shigella* in Bangladesh. Overall, this study predicts an abrupt decrease in the effectiveness of AZM against *Shigella* in the very near future and suggests prompt focus on seeking a more effective treatment alternative to AZM for shigellosis.

**KEYWORDS** *Shigella*, antimicrobial resistance mechanism, azithromycin, resistance gene cluster, whole genome

Address correspondence to Zhahirul Islam, zislam@icddrb.org.

The authors declare no conflict of interest.

[This article was published on 25 July 2022 without the equal-contribution note. The note was added in the current version, posted on 27 July 2022.]

*S*higella remains a foe in the public health sector, especially in resource-limited countries, due to its low infectious dose and the rise of multidrug resistance (MDR) phenomena (1–3). Annually, 188 million cases of dysentery occur globally due to *Shigella* infection, with 164,000 associated deaths (4). More than 98% of diarrheal deaths occur in low- and middle-income countries (LMICs) (1). The lack of treatment options and rapid propagation of antimicrobial resistance (AMR) factors make *Shigella* the leading cause of diarrheal death, especially among children in LMICs (1, 5–8). The ineffectiveness of previously used antibiotics, such as ampicillin, chloramphenicol, tetracycline, and sulfonamides, has coerced physicians to depend on a limited range of treatment options, like ciprofloxacin, ceftriaxone, and azithromycin (AZM) (5, 6, 9). AZM significantly lowers the persistence of shigellosis and is recommended for its effectiveness against *Shigella* infection (10, 11). However, recent global reports show a worsening scenario of decreased susceptibility to AZM in *Shigella* spp. (12–15). About 67% of *Shigella sonnei* isolates were reported to be AZM resistant in a recent observational study conducted in Bangladesh (11).

Several mechanisms decrease the efficacy of AZM in bacterial organisms, including harboring drug-modifying esterases and phosphotransferases, decreased permeability to antibiotics through changes in efflux pumps, target site modifications due to mutations (in 23S rRNA and two ribosomal proteins, L4 and L22, encoded by the *rplD* and *rplV* genes, respectively), and methylation of 23S rRNA (16). *Escherichia coli* has a higher susceptibility to acquire most of these AMR factors and is the closest pathovar to *Shigella* (17, 18). So, there is a high probability that other newer and/or novel mechanisms of AZM resistance remain to be identified in *Shigella*. Moreover, the mechanism of dissemination of the macrolide resistance factors from one species to another is of vast significance for a potential *Shigella* superbug. Conjugative R-plasmid-mediated horizontal gene transfer (HGT) has been demonstrated to be involved in the rapid transfer of genes responsible for resistance (19–22). Plasmid-mediated horizontal transfer of third-generation cephalosporins from *E. coli* to *Shigella* has already been reported in Bangladesh (23). Whole-genome sequencing (WGS) can be potentially informative for AMR mechanism studies. A large number of integrated tools and databases are available to identify AMR factors, plasmids, pathogenic gene islands, and resistance gene clusters from WGS data. Recently, a conjugative R-plasmid carrying the AZM resistance gene cassette [IS*26*-*mph*(A)-*mrx*(A)-*mph*(R)(A)-IS*6100*] named pKSR100 in *S. flexneri* serotype 3a was described to be involved in the intercontinental spread of AZM resistance among a men who have sex with men-associated outbreak lineage (19). Another cluster of *mph*(E)-*msr*(E)-IS*481* genes has been detected in the *Enterobacter hormaechei* plasmid pRIVM_C019595_1 (CP078057.1).

Although AZM is currently a worthwhile option for shigellosis treatment, the lack of efficacy studies and the absence of specific in-depth AZM shigellosis research leave this effective treatment option for shigellosis extremely vulnerable. Hence, intensive research on AZM resistance mechanisms in *Shigella* is warranted, especially in Bangladesh. Therefore, we report a comprehensive study that includes the identification of MRGs through WGS-based approaches to clearly depict the AZM resistance mechanisms and their rapid dissemination in *Shigella*.

## RESULTS

**Serological typing.** Among the 150 *Shigella* isolates tested in this study, *Shigella flexneri* was the predominant species (*n* = 83, 55.33%), followed by *Shigella sonnei* (*n* = 50, 33.33%), *Shigella boydii* (*n* =15, 10%), and *Shigella dysenteriae* (*n* = 2, 1.33%) (see Table S1 in the supplemental material). Within the *S. flexneri* isolates, serotype 2a was the most abundant (30.7%), followed by serotype 3a (10.7%), 4 (5.33%), 1c (2.7%), 6 (2%), and 3b (1.33%); the 1a, 1b, and 4b serotypes and atypical isolates each only comprised 0.7% of isolates. In *S. boydii*, the predominant serotype was *S. boydii* 2 (2.7%), followed by serotype 12 (2%), 8 (1.33%), and 1 (1.33%). Among the two *S. dysenteriae* isolates, the predominant serotype was serotype 2 (see Table S1).

**Antimicrobial susceptibility testing against AZM.** Thirty-eight percent (57/150) of the 150 *Shigella* isolates were found to be resistant to AZM in antimicrobial susceptibility tests. Among the four *Shigella* species, AZM resistance was most common in *S. sonnei*

**TABLE 1** Antimicrobial susceptibility test results for 150 *Shigella* isolates against AZM

| *Shigella* species | Total isolate count | Frequency of resistance phenotype [*n* (%)] | | Summary of MIC test (E-test) | | | |
| | | AZM$^r$ | AZM$^s$ | AZM$^r$ | | AZM$^s$ | |
| | | | | MIC (μg/mL) | Count | MIC (μg/mL) | Count |
|---|---|---|---|---|---|---|---|
| *S. flexneri* | 83 | 14 (17) | 69 (83) | >256 | 12 | 4 | 8 |
| | | | | 128 | 2 | 3 | 13 |
| | | | | | | 1 | 38 |
| | | | | | | 0.75 | 6 |
| | | | | | | 0.5 | 4 |
| Total or overall range | NA$^a$ | NA | NA | 128 to >256 | 14 | 0.5–4 | 69 |
| *S. sonnei* | 50 | 40 (80) | 10 (20) | >256 | 36 | 16 | 1 |
| | | | | 192 | 3 | 8 | 3 |
| | | | | 128 | 1 | 4 | 2 |
| | | | | | | 3 | 2 |
| | | | | | | 1 | 1 |
| | | | | | | 0.75 | 1 |
| Total or overall range | NA | NA | NA | 128 to >256 | 40 | 0.75–16 | 10 |
| *S. boydii* | 15 | 3 (20) | 12 (80) | >256 | 3 | 8 | 1 |
| | | | | | | 4 | 2 |
| | | | | | | 1 | 6 |
| | | | | | | 0.75 | 3 |
| Total or overall range | NA | NA | NA | >256 | 3 | 0.75–8 | 12 |
| *S. dysenteriae* | 2 | 0 (0) | 2 (100) | NA | NA | 3 | 1 |
| | | | | | | 0.75 | 1 |
| Total or overall range | NA | NA | NA | NA | NA | 0.75–3 | 2 |
| All *Shigella* species tested | 150 | 57 (38) | 93 (62) | 128 to >256 | 57 | 0.5–16 | 93 |

$^a$NA, not applicable.

(80%, 40/50), followed by *S. flexneri* (16.87%, 16/83), and *S. boydii* (20%, 3/15). Neither of the two *S. dysenteriae* isolates was resistant to AZM. The MIC determined through Epsilometer tests (E-test) ranged from 128 to >256 μg/mL for AZM-resistant *Shigella* isolates, whereas the MICs of AZM-sensitive *Shigella* isolates ranged from 0.5 to 16 μg/mL (Table 1). The MIC of AZM-sensitive *Shigella* isolates was 4 μg/mL, except for one *S. sonnei* isolate (16 μg/mL) and one *S. boydii* isolate (8 μg/mL) (Table 1). However, the majority of AZM-resistant isolates (51/57) exhibited MICs of >256 μg/mL.

**Identification of AZM resistance factors by PCR.** Among the 15 MRGs examined by PCR, the macrolide-2′-phosphotransferase [*mph*(A)] gene was present in all AZM-resistant *Shigella* isolates. The *erm*(B) (23S rRNA methylation) gene was detected in 42% (24/57) of the AZM-resistant *Shigella* isolates. Among the AZM-resistant isolates, 30% (12/40) of *S. sonnei*, 64% (9/14) of *S. flexneri*, and all 3 *S. boydii* isolates harbored the *erm* (B) gene (see Table S2). The *msr*(E) gene was found in two isolates, namely, *S. flexneri* Z12966 and *S. boydii* Z12959. PCR analysis of plasmid DNA and genomic DNA for the *mph*(A), *erm*(B), and *msr*(E) genes led to the same results (Table 2).

**AZM resistance factors in WGS analysis.** WGS analysis of the 13 *Shigella* isolates using AMRFinderPlus and the Resistance Gene Identifier (RGI) platform revealed another phosphotransferase gene, *mph*(E). All AZM resistance factors identified in the PCR study were substantiated by WGS analysis (Table 2). The *mph*(A) and *erm*(B) genes were concurrently found in all 13 AZM-resistant *Shigella* isolates subjected to WGS study. Additionally, the *mph*(E) and *msr*(E) genes coexisted in the AZM-resistant *S. flexneri* Z12966 and *S. boydii* Z12959 isolates. The multidrug efflux pump gene *emr*(E) was identified in all *S. flexneri* and *S. sonnei* isolates, the *acr*(F) gene was found in all *Shigella* spp. (except for *S. flexneri*), and the *mdt*(M) gene was only found in *S. boydii* isolates, regardless of AZM resistance (see Table S3).

**TABLE 2** Serotyping, AST, and PCR results for the 13 *Shigella* isolates subjected to WGS

| Strain | Serotype | PCR-identified AZM$^r$ factors | | AZM susceptibility test results | | |
| | | Genomic DNA | Plasmid DNA | Disc diffusion (mm) | E-test (mg/L) | AZM$^r$ status |
|---|---|---|---|---|---|---|
| *S. flexneri* strains | | | | | | |
| Z13145 | 2a | Absent$^a$ | Absent | 25 | 3 | AZM$^s$ |
| Z13032 | 2a | *mph*(A), *erm*(B) | *mph*(A), *erm*(B) | 7 | >256 | AZM$^r$ |
| Z12966 | 4 | *mph*(A), *ermB*(A), *msrE*(A) | *mph*(A), *erm*(B), *msr*(E) | 7 | >256 | AZM$^r$ |
| Z13164 | 3a | Absent | Absent | 22 | 4 | AZM$^s$ |
| K13242 | 3a | *mph*(A), *erm*(B) | *mph*(A), *erm*(B) | 7 | >256 | AZM$^r$ |
| *S. sonnei* strains | | | | | | |
| Z12947 | *S. sonnei*$^b$ | *mph*(A), *erm*(B) | *mph*(A), *erm*(B) | 7 | >256 | AZM$^r$ |
| Z12965 | *S. sonnei* | *mph*(A), *erm*(B) | *mph*(A), *erm*(B) | 7 | >256 | AZM$^r$ |
| Z13154 | *S. sonnei* | *mph*(A), *erm*(B) | *mph*(A), *erm*(B) | 7 | >256 | AZM$^r$ |
| Z13254 | *S. sonnei* | *mph*(A), *erm*(B) | *mph*(A), *erm*(B) | 7 | >256 | AZM$^r$ |
| *S. boydii* strains | | | | | | |
| Z12931 | 2 | *mph*(A), *erm*(B) | *mph*(A), *erm*(B) | 7 | >256 | AZM$^r$ |
| Z12959 | 3 | *mphA*(A), *ermB*(A), *msr*(E) | *mph*(A), *erm*(B), *msr*(E) | 7 | >256 | AZM$^r$ |
| Z12985 | 2 | Absent | Absent | 15 | 8 | AZM$^s$ |
| *S. dysenteriae* strain | | | | | | |
| Z12458 | 4 | Absent | Absent | 25 | 0.75 | AZM$^s$ |

$^a$AZM resistance factors were absent.
$^b$*S. sonnei* has one serotype only.

**Determination of mutation(s) in *rplD*, *rplV*, and 23S rRNA genes related to AZM resistance.** Analysis of the amplicon sequences of two rRNA protein genes (*rplD* and *rplV*) from the 63 *Shigella* isolates revealed no potential mutations or alterations that could be related to AZM resistance in *Shigella*. However, concomitant mutations at C495G (H165Q) in the *rp1D* (L4) gene and T137A (L46Q) and C12T in the *rp1V* (L22) gene were noted in all *S. flexneri* isolates. Another point mutation was identified at position C285T of the *rp1V* gene in the AZM-resistant Z12954 and Z13164 isolates. One allele of the 23S rRNA gene sequence was retrieved from each genome of the studied *Shigella* isolates. Seven alleles of the 23S rRNA gene were extracted from each of the four reference sequences, representing all four species of *Shigella*; the gene length was 2,904 bp in all isolates, except for that in *S. dysenteriae* (2,902 bp). None of the single-nucleotide polymorphisms or deletions present in the reported positions were associated with macrolide resistance among gut pathogens. Moreover, no single-nucleotide alteration was exclusively associated with AZM resistance.

**Genome BLAST distance phylogeny.** Whole-genome-based taxonomic clustering yielded nine species clusters (Fig. 1). The four species of *Shigella* could be separated into phylogenic clusters based on the respective reference sequences of *S. flexneri* ATCC 29903, *S. sonnei* ATCC 29930, *S. boydii* ATCC 8700, and *S. dysenteriae* ATCC 13313, which were obtained from the TYGS database (Fig. 1). The numbers above the branches in Fig. 1 represent genome BLAST distance phylogeny (GBDP) pseudo-bootstrap support values of >60% after 100 replications, with an average branch support of 65.3% (24). No specific clustering related to AZM susceptibility status was noted in the isolates in this study.

**Identification and analysis of plasmid sequences harboring MRGs.** PlasmidSPAdes was used to assemble the plasmid sequences with multiple contigs. PlasmidSeeker provides multiple clusters of reference plasmids present in the query sequence; each cluster contains multiple homologous reference plasmids and signifies the presence of a similar or homologous plasmid in the query sequence. The IncFII-type plasmid was most common and was present (i.e., 95.79 to 100.0% identity) in all isolates, except for the *S. sonnei* Z12965 and *S. sonnei* Z13154 isolates. The IncB/O/K/Z-type plasmid was the second most common (94.08 to 100.0% identity). The Col(BS512)-type and Col156-type plasmids were present in all *S. boydii* isolates (see Table S3). The pKSR100-like plasmid was identified in three isolates (*S. flexneri* Z12966, *S. flexneri* Z13032, and one

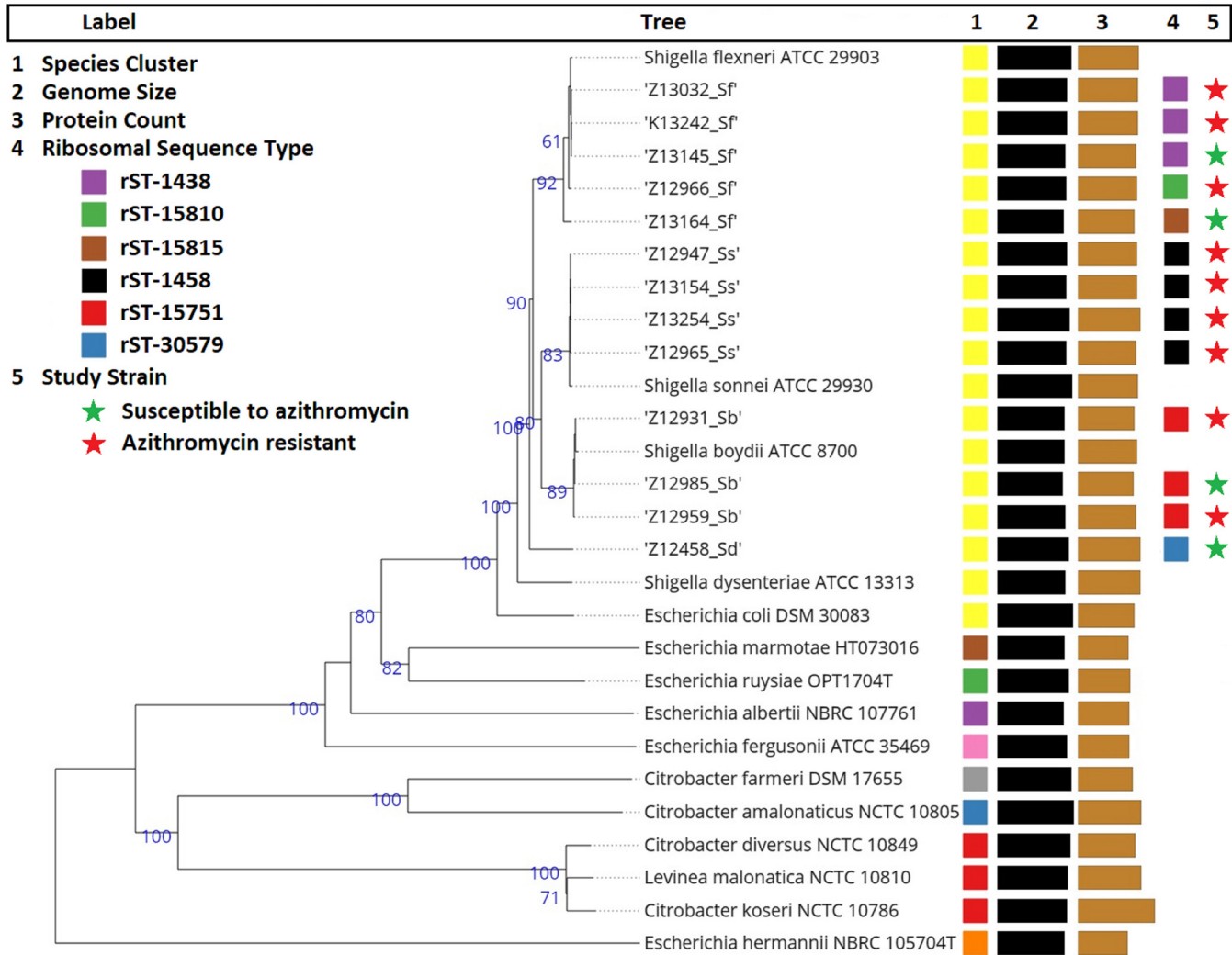

**FIG 1** Tree inferred with FastME 2.1.6.1 based on GBDP distances calculated from genome sequences. The tree was rooted at the midpoint. Values in the cluster nodes represents pseudo-bootstrap values (≥60). Species, genome size, protein count, and ribosomal sequence type are represented with multicolor blocks. Study strains are denoted with asterisks; green and red asterisks represent AZM-sensitive and -resistant *Shigella* isolates, respectively.

*S. boydii* Z12959 isolate) based on NCBI BLAST searches followed by Gview alignment of the plasmid sequences. Multiple contigs of the whole *S. flexneri* Z13032 genome showed ≥90% identity to pKSR100. A 69,536-bp fragment (98.14% identity) and 6,362-bp fragment (99.74% identity) covered 94% of the pKSR100 plasmid in *S. flexneri* Z13032. We named the *de novo*-assembled pKSR100-type plasmids in *S. flexneri* Z13032, *S. flexneri* Z12966, and *S. boydii* Z12959 pZ13032_1, pZ12966_1, and pZ12959_1, respectively (Fig. 2). Annotation data for the plasmid sequences indicated the presence of MRGs in the IS*26-mph*(A)-*mrx*(A)-*mph*(R)(A)-IS*6100* resistance gene cluster in the pZ12959_1, pZ12966_1, and pZ13032_1 plasmids. The *erm*(B) gene was found to neighbor the IS*26-mph*(A)-*mrx*(A)-*mph*(R)(A)-IS*6100* cluster in all three plasmids (Fig. 2). The other two MRGs, *msr*(E) and *mph*(E), were found contiguously in a 4,952-bp sequence in both *S. boydii* Z12959 (JAFEJL010000224.1) and *S. flexneri* Z12966 (JAEUXL010000191.1), where these MRGs were arrayed as an *mph*(E)-*msr*(E)-IS*481*-IS*6* gene cluster.

## DISCUSSION

This study intended to draw an all-inclusive map of the mechanisms of AZM resistance in *Shigella* spp. in Bangladesh. We found resistance to AZM was primarily

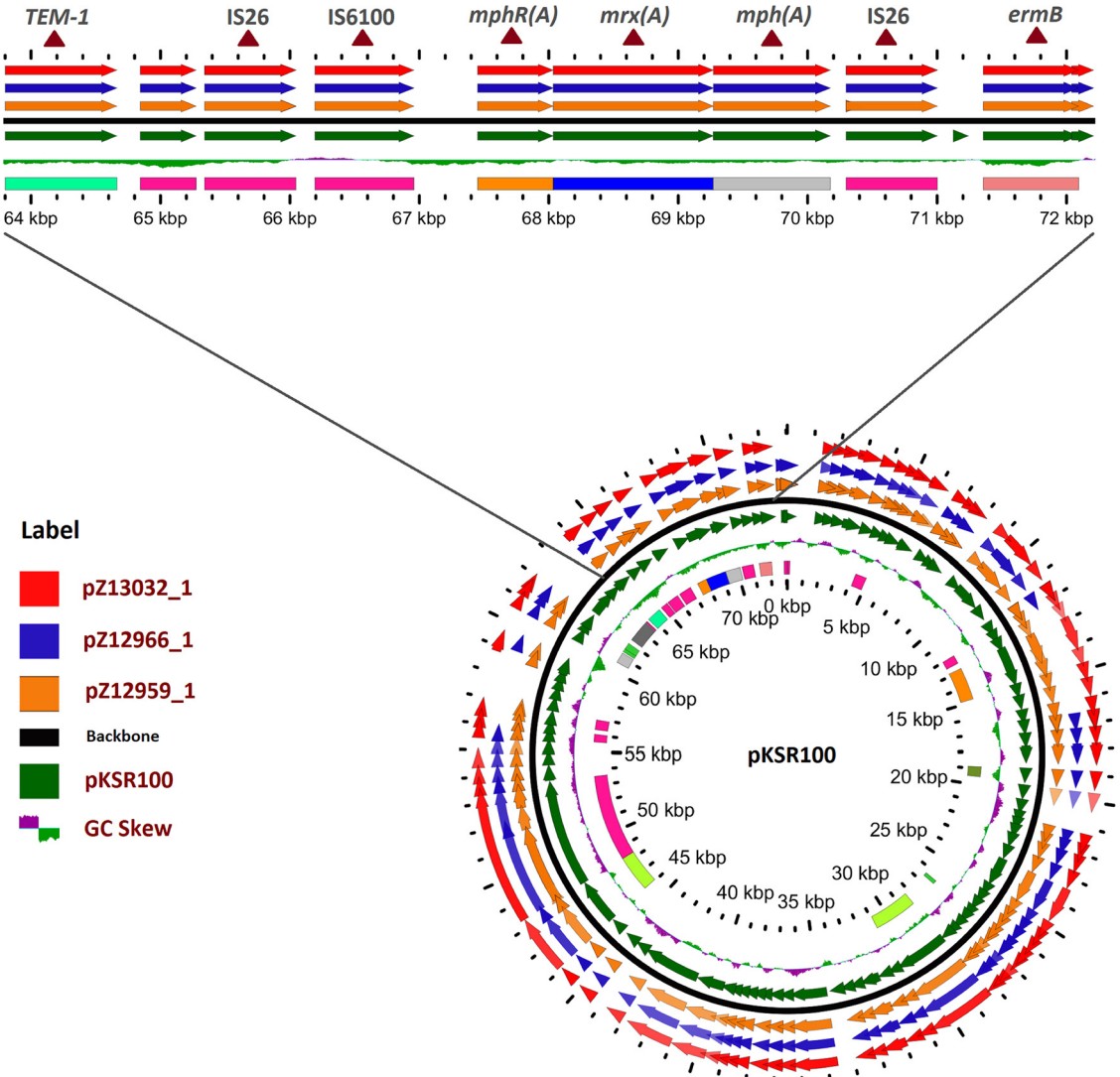

**FIG 2** Comparison (BLAST atlas) of the plasmids of *Shigella* identified using the Gview server. The innermost multicolor ring shows the Clusters of Orthologous Groups (COG) followed by GC skews, ring backbone, pKSR100, and pZ12959_1, pZ12966_1, and pZ13032_1 in the outermost ring. Zoomed regions show AZM resistance-related gene clusters in the plasmids.

encoded by the plasmid-mediated genes *mph*(A) (100%) and *erm*(B) (24%). To the best of our knowledge, this is the first report of the *msr*(E) and *mph*(E) MRGs expediting resistance to AZM in *Shigella* in Bangladesh. All four MRGs were embedded in plasmid-borne resistance gene cassettes and neighbored by multiple IS*6* family insertion sequences. Furthermore, AZM resistance was significantly more frequent in *S. sonnei* than in the other three *Shigella* spp., which has probably contributed to the epidemiological drift of *S. sonnei* over *S. flexneri* in Bangladesh. Overall, AZM resistance in *Shigella* was found to be mediated by multiple AMR mechanisms, including drug modification [phosphotransferase genes *mph*(A) and *mph*(E)], 23s rRNA methylation [*erm*(B) gene], and inhibition of the activation of an efflux pump [*msr*(E) gene]. In addition, we found these MRGs are rapidly disseminating via HGT of plasmid-embedded macrolide resistance gene cassettes from one bacterium to another.

We combined a range of traditional AMR screening strategies, amplicon (Sanger) sequencing of ribosomal protein genes, and whole-genome-based techniques to strengthen the investigation process. This study is the first to employ such an elaborate approach to *Shigella* spp. in Bangladesh and provided a substantial overview of AZM

resistance mechanisms in this population of bacteria. Several analyses of specific macrolide resistance genes or mechanisms and two WGS studies conducted in Europe, the United States, and Asia have reported some of the mechanisms of AZM resistance in *Shigella* (14, 19, 25–28).

The emergence of the invasive *mph*(A) gene in *Shigella* has long been postulated by several studies from Europe, the United States, and Asia (12, 14, 15). One study in Bangladesh reported the presence of the *mph*(A) gene as one mechanism of AZM resistance in *Shigella* (15). Our investigation conveys a clear message that the *mph*(A) gene appears to play a key role in the rising AZM resistance among *Shigella* in Bangladesh. The major methylation gene *erm*(B) methylates an adenine at position A2058 of the 23S rRNA gene, which modifies the ribosomal structure and alters drug-target affinity, which in turn generates resistance against macrolides (29, 30). The contribution of the *erm*(B) gene to AZM resistance in *Shigella* is not new knowledge; however, the frequency of this gene in the studied population seems extremely high (24%), especially in *S. flexneri* (60%) (14, 31, 32). The higher incidence of *erm*(B) in the study isolates may be due to the presence of this gene at a position adjacent to the highly invasive IS*26*-*mph*(A)-*mrx*-*mph*(R)(E)-IS*6100* cluster, as identified in pKSR100 orthologous plasmids. The novel incorporation of *msr*(E) and *mph*(E) genes in *Shigella* is not unexpected, as these genes have been frequently reported in *E. coli* and other *Enterobacteriaceae* (14, 19, 33, 34). In the current study, these two MRGs [*msr*(E) and *mph*(E)] were found to be integrated into an *mph*(E)-*msr*(E)-IS*482*-IS*6* gene cassette, possibly through plasmid-mediated transfer.

The Sanger sequencing and whole genome-based approaches revealed some concomitant mutations in both ribosomal protein genes (*rplD* and *rplV*) in all *S. flexneri* isolates, regardless of their resistance to AZM. Punctual mutations and insertions and deletions in two ribosomal proteins (L4, encoded by the *rplD* gene, and L22, encoded by the *rplV* gene) affect the MICs of macrolides by reducing the binding affinity; thus, they induce macrolide resistance (16, 35–38). However, the absence or ambiguous presence of punctual mutations in the *rplD* and the *rplV* genes in AZM-resistant and -susceptible *Shigella* isolates in this study signifies that these mutations have no involvement in AZM resistance. Moreover, our whole genome-based bioinformatics approach to evaluate the role of 23S rRNA gene mutation(s) also provided inconclusive insights to define AZM resistance in *Shigella*.

WGS and plasmid analysis demonstrated that the presence of AZM resistance determinants were affiliated with the AMR mosaic regions IS*26*-*mphA*-*mrx*-*mphR*(A)-IS*6100* and *mph*(E)-*msr*(E)-IS*482*-IS*6*. The pathogenic IS*26*-*mph*(A)-*mph*(R)(A)-IS*6100* gene cluster was previously described in the pKSR100 plasmid, confers high-level resistance to AZM, and was detected in various intercontinental disseminated sublineages of *S. flexneri* 3a (19). The other AMR gene cluster, *mph*(E)-*msr*(E)-IS*482*-IS*6*, was identified for the first time in *Shigella* isolates. The coexistence and critical roles of the *msr*(E) and *mph*(E) genes in macrolide resistance in clinical *Pseudomonas aeruginosa* isolates was previously described (39, 40). The plasmid-borne *mph*(E)-*msr*(E)-IS*482*-IS*6* gene cluster was also found in the *E. coli* plasmid pRIVM_C019595_1 (NZ_CP068889.1) with > 99% identity and in *Enterobacter hormaechei* strain EH_316 plasmid pEH_316-2 (NZ_CP078057.1) with 100% identity. Thus, the isolated 4292 bp fragment [*mph*(E)-*msr*(E)-IS*482*-IS*6* gene cluster) could be a small part of a plasmid or a transposon-like MGE that was found as separate contig in the studied genomes. Multiple IS*6* family insertion sequences were observed both upstream and downstream of the AMR genes in both gene clusters. These IS*6* family insertion sequences in the plasmid-borne pathogenic gene clusters may play a crucial role in strengthening the invasiveness and invasion capability and thus facilitate AZM resistance in *Shigella* (41).

Antimicrobial resistance mechanisms are extremely perplexing; thus, it is difficult to draw definitive conclusions regarding the complex mechanisms of AZM resistance in *Shigella*. Plasmid sequencing and sequencing of all alleles of the *23S rRNA* gene may add more strength to this study. However, the multiple potential

macrolide resistance mechanisms identified in this work, particularly the possibility of their transmission as plasmid-integrated resistant gene cassettes, provide a stark warning of the high risk of losing this crucial drug from the very short list of treatment options for shigellosis. Hence, this study highlights the need for new alternatives to fight shigellosis in countries such as Bangladesh, where *Shigella* is endemic. This study also recommends surveillance studies should be initiated to track the intra- and interspecies horizontal spread of the newly evolved MRGs both locally and internationally.

## MATERIALS AND METHODS

**Bacterial strain isolation.** A total of 150 *Shigella* isolates were isolated between 2016 and 2018 in the Laboratory of Gut-Brain Signaling, Laboratory Sciences and Services Division, icddr,b. All *Shigella* isolates were confirmed by standard biochemical procedures and serotyped by slide agglutination tests using commercial monoclonal antisera (Denka Seiken, Tokyo, Japan) (42). The isolates were stored at −80°C for further investigation.

**Antimicrobial susceptibility tests.** As there are no clinical breakpoints (MIC or disc diffusion test diameter) to define AZM resistance in *Shigella*, the epidemiological cutoff values defined by the Clinical and Laboratory Standards Institute (CLSI) for *S. flexneri* and *S. sonnei* were used in this study (43). The disc diffusion method was performed using Mueller-Hinton agar and commercially available 15-$\mu$g AZM antibiotic discs (Oxoid, Basingstoke, United Kingdom) according to the CLSI guidelines for *Enterobacteriaceae* (43). *E. coli* ATCC 25922 was used as a control for the antimicrobial susceptibility tests. The MIC was determined by epsilometer tests using Etest strips (Liofilchem, TE, Italy).

**Genomic DNA and plasmid DNA extraction.** All 63 *Shigella* isolates, including 57 AZM-resistant and 6 AZM-sensitive isolates, were enriched overnight in Luria-Bertani broth at 37°C. Genomic DNA was extracted using Wizard Genomic DNA purification kits (Promega, Madison, WI, USA), and the PureYield plasmid miniprep kit system (Promega) was used to extract plasmid DNA from the selected isolates. The purity and quantity of the extracted DNA samples were checked using a NanoDrop spectrophotometer (Thermo-Scientific, USA). All DNA and plasmid samples were stored at −20°C and 4°C, respectively, until analysis.

**PCR amplification of antibiotic resistance genes.** Genomic DNA was extracted from the 57 AZM-resistant and 6 AZM-sensitive *Shigella* isolates and subjected to PCR to identify the presence of 15 macrolide resistance genes: the methylase-encoding genes *erm*(A), *erm*(B), *erm*(C), *erm*(F), and *erm*(42); the esterase-encoding genes *ere*(A) and *ere*(B); the phosphotransferase-encoding genes *mph*(A), *mph*(B), and *mph*(D); and the transferable efflux system-encoding genes *msr*(A), *msr*(D), *msr*(E), *mef*(A), and *mef*(B). The primer sequences, amplicon size, and annealing temperatures are presented in Table S4 in the supplemental material. Plasmid DNA samples were also used as PCR templates to identify the *mph*(A), *erm*(B), and *msr*(E) genes in the isolated plasmids.

**Determination of AZM resistance-related mutation(s) in the *rplD*, *rplV*, and *23S rRNA* genes.** The *rplD* and *rplV* genes encoding the ribosomal proteins L4 and L22, respectively, were sequenced using an *rplD* primer (5′-TGC TGC TGG TTA AAG GTG CTG TCC C) and *rplV* primer (5′-GGT GAA TTC GCA CCG ACT CGT ACT TAT CG) (36) with ABI PRISM BigDye Terminator cycle sequencing reaction kits (Applied Biosystems, Foster City, CA, USA) on a Genetic Analyzer 3500 XL (Thermo Fisher Scientific). The *23S* rRNA gene sequences were extracted from 13 whole genomes and 4 reference sequences (*S. flexneri* ATCC 29903 [CP026788.1], *S. sonnei* ATCC 29930 [CP026802.1], *S. boydii* ATCC 8700 [CP026731.1], and *S. dysenteriae* ATCC 13313 [CP026774.1]) using barrnap v0.9 tools and applying the –outseq flag with the default settings (44). The *23S* rRNA gene sequences from *S. flexneri* 2a strain 301 (NR-076170.1), *S. sonnei* strain Ss046 (NR_076358.1), *S. boydii* Ss227 (NR_076357.1), and *S. dysenteriae* Sd197 (NR_076356.1) were obtained from the NCBI GenBank database. All of the extracted and web-retrieved sequences were aligned and analyzed using MEGA X v10.2.6 software (45).

**Whole-genome sequencing.** Nine AZM-resistant and four AZM-sensitive *Shigella* isolates were selected for whole-genome sequencing. These isolates were sorted based on the presence of the *mph*(A) and *erm*(B) MRGs, which were confirmed by PCR. The selected isolates included four *S. sonnei* isolates, three *S. boydii* isolates from two serotypes (2 and 3), five *S. flexneri* isolates from three serotypes (2a, 3a, and 4), and one *S. dysenteriae* isolate (Table 1). All *Shigella* isolates were enriched in Luria-Bertani broth at 37°C for 16 h, and genomic DNA was extracted using QIAamp DNA minikits (Qiagen) according to the manufacturer's instructions. Whole-genome sequencing was carried out using Illumina technology at the Genomics Centre of the International Centre for Diarrheal Disease Research, Bangladesh (icddr,b), and the sequences were processed and assembled using a previously described methodology (46). The obtained genome sequences have been published in GenBank under the BioProject accession numbers PRJNA693631, PRJNA694802, PRJNA698772, PRJNA704496, and PRJNA698078 and announced publicly (46). Annotation of the genomes was performed by NCBI through the NCBI Prokaryotic Genome Annotation Pipeline v5.0 (47).

**Sequence typing and phylogenetic analysis.** The genome sequence data were uploaded to the Type (Strain) Genome Server (TYGS), a free bioinformatics platform available at https://tygs.dsmz.de, for whole-genome-based taxonomic analysis (48). The nearest genome typing of the isolates was performed by comparing the query genomes against all available strain type genomes available in the TYGS database via an approximation of intergenomic relatedness (MASH) algorithm (49). The 10 strain types with the smallest MASH distances were chosen per query genome. The resulting intergenomic distances were

used to deduce a balanced minimum evolution tree with branch support (from 100 pseudobootstrap replicates) via FASTME 2.1.6.1 including later SPR processing (50). The trees were rooted at the midpoint (51) and visualized using PhyD3 (52). The branch lengths were scaled in terms of GBDP distance formula d5 and the GreedyWithTrimming distance algorithm. Type-based species clustering using a 70% digital DNA-DNA hybridization (dDDH) radius around each of the 15 types of isolates was conducted as described previously (48). Subspecies clustering was performed by setting a 79% dDDH threshold, as described previously (53).

**Identification of AZM resistance genes in whole-genome sequences.** AMR and virulence genotypes were identified with NCBI's pathogen detection pipeline tool AMRFinderPlus v3.10.5 and the web portal of the Resistance Gene Identifier (RGI) v5.2.0 provided in the Comprehensive Antibiotic Resistance Database v3.1.3 (54, 55). The NCBI-curated Reference Gene Database and a curated collection of hidden Markov models were used in AMRFinderPlus. In RGI, perfect and strict hits were identified by excluding nudging loose hits with ≤95% identity.

**Plasmid sequence identification and typing.** Plasmid sequences were determined using multiple *in silico* approaches. PlasmidSPAdes (v3.13.0) and PlasmidSeeker v1.3 were used to identify the plasmids from the FASTQ reads (paired end, R1 and R2) (56, 57). PlasmidSPAdes assembly resulted in plasmid sequences with multiple contigs. PlasmidSeeker provided multiple clusters of reference plasmids that were present in the query sequences. The assembled plasmid sequences and whole-genome sequences were then used as target sequences in the nucleotide BLAST searches (http://blast.ncbi.nlm.nih.gov/Blast .cgi) by using the pKSR100 reference sequence as the query. The pangenome and BLAST atlas were constructed using the Gview tool (https://server.gview.ca/); pKSR100 was used as the reference core for the BLAST atlas module (58). Plasmid sequences were typed using PlasmidFinder v2.1 (https://cge.cbs.dtu.dk/ services/PlasmidFinder/), maintaining 90% threshold identity and 60% minimum coverage (59).

**Data availability.** All WGS data in this study have been submitted to NCBI GenBank and are available under the BioProject numbers PRJNA693631, PRJNA694802, PRJNA698078, and PRJNA698772. The accession links to the reference sequences used in this study are hyperlinked. All raw data and primary outputs of genome sequence analyses are available from the corresponding author upon reasonable quest.

## SUPPLEMENTAL MATERIAL

Supplemental material is available online only.
**SUPPLEMENTAL FILE 1**, XLSX file, 0.04 MB.

## ACKNOWLEDGMENTS

This research activity was funded by icddr,b, Dhaka, Bangladesh. Z.I. received grant support from the Fogarty International Center, National Institute of Neurological Disorders and Stroke of the National Institutes of Health, USA, under award number K43TW011447. S.N. received grant support from East West University, Dhaka, Bangladesh, under award number of EWCRT (RP)-R10(4)-2017-OL(2). S.H. received grant support from the Global Health Equity Scholars NIH FIC TW010540, USA. We acknowledge Shah Nayeem Faruque for his contribution in the critical review of the manuscript. Icddr,b gratefully acknowledges the commitment of the Government of Bangladesh to its research efforts and also acknowledges with gratitude the governments of Canada, Sweden, and the UK for their unrestricted support.

The study was reviewed and approved by the Institutional Review Board and the Ethical Committee of icddr,b, Dhaka, Bangladesh.

Z.I. and S.N. conceptualized the study with their expertise. The research methodology and execution plan were devised by A.A. and S.N. Z.I. and S.N. contributed to the acquisition of funding. Material acquisition, technique optimization, laboratory experiments, and data acquisition were performed by A.A., B.I., R.B., and F.H.N. Statistical analysis and interpretations were conducted by A.A. and S.N. Genomic data analysis, visualization, and interpretation were performed by A.A. Results were investigated and inspected by A.A., S.N., and S.H. Z.I. supervised the whole study. S.N. and A.A. drafted the primary manuscript, which was exclusively scrutinized by Z.I. and S.H. Proofreading and reviews were done by all other authors. All authors read and approved the final manuscript before submission.

We declare no competing interests.

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
