## [Reviewer comments · Microbiology Spectrum]

Microbiology Spectrum

Multiple Mechanisms Confer Resistance to Azithromycin in *Shigella* in Bangladesh: A Comprehensive Whole Genome-Based Approach

Suraia Nusrin, Asaduzzaman Asad, Shoma Hayat, Bitali Islam, Ruma Begum, Fahmida Nabila, and Zhahirul Islam

Corresponding Author(s): Zhahirul Islam, International Centre for Diarrhoeal Disease Research

Review Timeline:

Submission Date:	March 6, 2022
Editorial Decision:	April 20, 2022
Revision Received:	June 19, 2022
Accepted:	June 27, 2022

Editor: Monica Garcia-Solache

Reviewer(s): Disclosure of reviewer identity is with reference to reviewer comments included in decision letter(s). The following individuals involved in review of your submission have agreed to reveal their identity: raju_yadav Yadav (Reviewer #1); Chien-Shun Chiou (Reviewer #2)

Transaction Report:

DOI: <https://doi.org/10.1128/spectrum.00741-22>

April 20, 2022

Dr. Zahirul Islam
International Centre for Diarrhoeal Disease Research
Laboratory of Gut-Brain Signaling
68. Tajuddin sarani
Dhaka 1212
Bangladesh

Re: Spectrum00741-22 (Multiple Resistance Mechanisms Conferring Reduced Susceptibility to Azithromycin in *Shigella* in Bangladesh: A Whole Genome Based Comprehensive Approach)

Dear Dr. Zahirul Islam:

Before the manuscript can be accepted it will need to undergo modifications as suggested by reviewer 2. Additionally I have several comments that I would like to see addressed:

Why do the authors consider "reduced susceptibility to azithromycin" isolates as non wild type? Are these isolates not coming from patients, have these isolates been modified in any way in the laboratory? Otherwise even if with an altered resistance profile are wild type.

- Table 1 is confusing, does the "susceptibility to azithromycin" columns means disc diffusion size? Please clarify.
- Colored blocks in figure 1 are not easy to understand and do not provide too useful information, I suggest to simplify the figure. What does "user strain" means, the authors have that label even for ATCC strains in figure 5 such as Sb ATCC8700, please clarify.
- Figure 2: as reviewer 2 notices, the authors should state if the sequence of the plasmids was fully determined and how they verified it was so.
- Figures 2 and 3 are a bit redundant and could be simplified in a single figure.
- Figure 4 is confusing and perhaps unnecessary.
- I would suggest to move table 3 to supplementary materials.
- Supplementary figure 1 is confusing and redundant with figure 5

Please put references in a uniform format, some have fully spelled authors last names and other only the initials.

Link Not Available

Sincerely,

Monica Garcia-Solache

Journals Department
Reviewer comments:

Reviewer #1 (Comments for the Author):

Abstract language should be standard.

Reviewer #2 (Comments for the Author):

The authors determined azithromycin resistance determinants in 57 of 150 *Shigella* isolates collected in Bangladesh between 2016 and 2018. Five resistance genes *mphA*, *emrB*, *msrE*, *mphE*, and *ermE* were identified in the RSA isolates, but the chromosome-borne *ermE* did not confer resistance to azithromycin. The manuscript is too long and several paragraphs and figures/tables are not relevant to azithromycin resistance. Therefore, the manuscript is not strictly focused.

Major concern:

1. What do the 150 isolates represent? A country, or a certain area of the country?
2. There were 80 % (40/50) of *S. sonnei* isolates resistant to azithromycin. Were the RSA isolates clonal? Were they collected from a common outbreak? Were they derived from a recent common ancestor?

Minor concern:

1. Line 39, "All four MRGs were found to be plasmid-borne and belong to the IncFII family" What data or experiments support this conclusion?
2. Table S2. The total number of isolates is 149. The number of *Shigella dysenteriae* 2 should be 2. And, the values in the column of "Percentage" should have the same decimal numbers.
3. Line 100, Among the 150 *Shigella* "strains"... "isolate(s) rather than "strain(s)" should be used in most circumstances.
4. Line 102, "Supplementary table S2" appears first in the text, it should be revised as table S1.
5. Table S1. Suggest adding the sizes of amplicons in the Table.
6. Line 121, "Azithromycin" Should be replaced by "azithromycin"
7. All the supplementary tables (S1, S2, S3) and Supplementary file 1 can be put in an excel file in different spreadsheets.
8. Line 125, ...(Supplementary file S1). In the excel file, it is marked as "Supplementary file S2".
9. Lines 126-127, "PCR of plasmid DNA and genomic DNA for *mphA*, *ermB* and *msrE* genes exhibited the same results" It is confusing! How did the authors separate plasmid DNA from genomic DNA?
10. Lines 134-135, "Moreover, the *mphA* and *ermB* genes were concurrently found in all RSA *Shigella* strains" All the RSA isolates selected for WGS are PCR positive for *mphA* and *ermB*, aren't they?
11. Lines 149-157 "Genome BLAST Distance Phylogeny (GBDP)" Is this study relevant to azithromycin resistance?
12. Lines 175-177, "The putative plasmid sequences extracted from *S. flexneri* Z12966 and *S. flexneri* Z13032 were named as pZ12966_11 and pZ13032 respectively (Figure 2)" Without long sequence reads, how did the authors fill the gaps of fragments to obtain complete plasmid sequences? How did the authors make sure the fragments belonged to plasmids instead of chromosomes?
13. The genetic maps in Figure 2 and Figure 3 are not consistent. Suggest simplifying the maps.
14. Lines 191-192, "These sequences were named as and pZ12966_191 respectively" The 4,952 bp segment should be only part of a plasmid, it should not be named as a plasmid ppZ12959_224 (pZ12966_191).
15. Figure 4 is not necessary, it should be removed.
16. Figure 5 and Figure S1. Are they relevant to azithromycin resistance?
17. Lines 196-215, This paragraph is not relevant to azithromycin resistance, it (and Table S3) should be excluded.

Staff Comments:

Preparing Revision Guidelines

Please return the manuscript within 60 days; if you cannot complete the modification within this time period, please contact me. If you do not wish to modify the manuscript and prefer to submit it to another journal, please notify me of your decision immediately so that the manuscript may be formally withdrawn from consideration by Microbiology Spectrum.

Articles review comments:

In abstract language need to be correct, also the genomic sequence mention must have the reference and also the proper function of the genomic sequence is not mention.

Rest is acceptable...

Re: Spectrum00741-22 (Multiple Mechanisms Conferring Resistance to Azithromycin in *Shigella* in Bangladesh: A Whole Genome Based Comprehensive Approach)

Response to Editor's Comments:

Thanks a lot for your concern regarding the study and your kindly comments. We carefully consider each of yours and reviewers suggestions and tried our best to improve the revised manuscript accordingly. Please allow us to respond all your questions separately as follows.

Comment: *Why do the authors consider "reduced susceptibility to azithromycin" isolates a non-wild type? Are these isolates not coming from patients, have these isolates been modified in any way in the laboratory? Otherwise even if with an altered resistance profile are wild type.*

Response: Thanks for your comments. We are pleased to inform you that all of the isolates were isolated from fecal specimens of diarrhoeal patients admitted in Dhaka Hospital of icddr,b. We characterized all these strains without any modification in the laboratory and we agree with editor that strain with an altered resistance profile are wild type. We used the term "non-wild-type" to describe the reduced susceptibility to macrolide in *Shigella* strains as per current Clinical and Laboratory Standards Institute (CLSI) guideline (CLSI-2016: Page 60, Table 2A-2 (page: 60) and Appendix G (page: 238). It describes the strains with MIC values below the epidemiological cutoff values (ECV) (presumed not to possess resistance mechanisms) as wild type" (WT) and the strains with MIC values above the ECV (presumed to possess resistance mechanisms) as "non-wild-type" (NWT). However, in response to the editor's comment, we simply described strains as resistant and sensitive in the current version of the manuscript.

Reference: CLSI. *Performance Standards for Antimicrobial Susceptibility Testing. 26th ed. CLSI supplement M100S. Wayne, PA: Clinical and Laboratory Standards Institute; 2016.*

Comment: *Table 1 is confusing, does the "susceptibility to azithromycin" columns means disc diffusion size? Please clarify.*

Response: Thanks for your comment. We apologize for the inconvenience. The decisive summary (azithromycin resistant/sensitive) of antibiotic susceptibility tests (both disc diffusion test and MIC test) was placed under the "susceptibility to azithromycin" column of figure 1. We changed the column header as "Frequency of AZM^R /AZM^S strains" in place of "susceptibility to azithromycin" in the current version of the manuscript (Figure 1, column 3). The detailed results are also available in the supplementary table S2.

Comment: *Colored blocks in figure 1 are not easy to understand and do not provide too useful information, I suggest to simplify the figure.*

Response: Thanks for your suggestion. We modified the figure 1 in the revised version to make it simplified, understandable and more informative.

Comment: *What does "user strain" means, the authors have that label even for ATCC strains in figure 5 such as Sb ATCC8700, please clarify.*

Response: Thanks for your comments. The "user strain" was denoting the *Shigella spp.* isolated, sequenced and analyzed in the current study. In the figure 5 (previous), the study strains were denoted with "*" sign and no ATCC strains were marked as user strain. We remove the figure 5 in the revised version of the manuscript in response to your comments and the reviewer#2's suggestion.

Comments: *Figure 2: as reviewer 2 notices, the authors should state if the sequence of the plasmids was fully determined and how they verified it was so.*

Response: Thank you for your valuable instruction. We determined whole plasmid sequences with multiple contigs which have $\geq 94\%$ coverage to the reference plasmids of interest. We addressed the comments from reviewer 2 and incorporated the detail procedure in the revised version of the manuscript in methodology section (page 17-18; line 354 – 365) as follows-

“Plasmid sequences were determined using multiple in-silico approaches. PlasmidSPAdes (SPAdes v3.13.0) and PlasmidSeeker v1.3 were used to determine the plasmids from the FASTQ reads (Paired end, R1 and R2). PlasmidSPAdes assembly resulted in plasmid sequences in multiple contigs. PlasmidSeeker provided multiple clusters of reference plasmids that supposed present in the query sequence. The assembled plasmid sequences and whole genome sequences were then used as target sequences in the nucleotide blast (<http://blast.ncbi.nlm.nih.gov/Blast.cgi>) study with reference query sequence of **pKSR100**. Pangenome and blast atlas were constructed using the Gview tool (<https://server.gview.ca/>) where **pKSR100** was used as the reference core for the blast atlas module. Plasmid sequences were typed by PlasmidFinder v2.1 (<https://cge.cbs.dtu.dk/services/PlasmidFinder/>) maintaining 90% threshold identity and 60% minimum coverage.”

Comments: *Figures 2 and 3 are a bit redundant and could be simplified in a single figure.*

Response: Thanks for the suggestion. We merged the figure 2 and figure 3 as single figure (figure 2) and incorporate as figure 2 in the revised version of the manuscript.

Comments: *Figure 4 is confusing and perhaps unnecessary.*

Response: Thanks for your suggestion. Figure 4 was deleted in the current version of the manuscript.

Comments: *I would suggest to move table 3 to supplementary materials.*

Response: Thanks for your suggestion. We moved the table 3 in the supplementary file.

Comments: *Supplementary figure 1 is confusing and redundant with figure 5*

Response: Thanks for your comment. As per your concern and suggestion from reviewer#2, we removed the Figure 5 and supplementary figure S1 from revised version of the manuscript.

Comments: *Please put references in a uniform format, some have fully spelled authors last names and other only the initials.*

Response: Thanks for your keen observation. We apology for unintentional error; but, we carefully checked and corrected all the references as per the journal's guidelines in the current version of the manuscript.

Reviewer comments:

Reviewer #1 (Comments for the Author):

Abstract language should be standard.

Response: Thank you for your constructive comment. Current version of the manuscript has been edited with native English language editor (page: 2-3; Line: 25-56).

Reviewer#2(Comments for the Author):

Overall: The authors determined azithromycin resistance determinants in 57 of 150 *Shigella* isolates collected in Bangladesh between 2016 and 2018. Five resistance genes *mphA*, *emrB*, *msrE*, *mphE*, and *ermE* were identified in the RSA isolates, but the chromosome-borne *ermE* did not confer resistance to azithromycin. The manuscript is too long and several paragraphs and figures/tables are not relevant to azithromycin resistance. Therefore, the manuscript is not strictly focused.

Response: Thanks a lot for your constructive criticism to improve the revised version of our manuscript. We carefully considered each of your suggestions in the current version of this manuscript.

We identified four macrolide resistance genes, *mphA*, *ermB*, *msrE* and *mphE* that were associated with azithromycin resistance. We also identified 3 chromosomal genes, *emrE*, *mdtM* and *acrF* in this study that were associated with multidrug efflux system. These drug efflux systems could recognize multiple antibiotics and many other compounds like macrolides (Nishino et al., 2021). As per editor and reviewer suggestion, we reduced several paragraphs and figures in the revised version of the manuscript as per our study objectives.

Reference: Nishino K, Yamasaki S, Nakashima R, Zwama M, Hayashi-Nishino M. Function and Inhibitory Mechanisms of Multidrug Efflux Pumps. *Front Microbiol.* 2021 Dec 3;12:737288. doi: 10.3389/fmicb.2021.737288. PMID: 34925258; PMCID: PMC8678522.

Major concern:

Comment 1: What do the 150 isolates represent? A country, or a certain area of the country?

Response: icddr,b is an international health research institution based in Bangladesh. icddr,b Dhaka Hospital is the world's largest diarrhoeal disease hospital. Considering geographical location of this hospital, most of the shigellosis patients diagnosed elsewhere in Bangladesh referred to icddr,b's hospital. Therefore, one hundred and fifty isolates of *Shigella* represent the country during the isolation period.

Comment 2: There were 80 % (40/50) of *S. sonnei* isolates resistant to azithromycin. Were the RSA isolates clonal? Were they collected from a common outbreak? Were they derived from a recent common ancestor?

Response: Thanks for your comments. All the isolates (n=150) of *Shigella* spp. collected from the sporadic cases of Shigellosis during the study period. These isolates were not related with any outbreak of Shigellosis. Our molecular typing data also suggested that these isolates were not clonal.

Minor concern:

Comment 1. Line 39, "All four MRGs were found to be plasmid-borne and belong to the IncFII family" What data or experiments support this conclusion?

Response: Thanks for your nice comments on MRGs. We would like to confirm you that we identified three MRGs, namely *mphA*, *ermB*, and *msrE* from plasmid DNA. Further, annotation data of the identified pKSR100 (IncFII type plasmid) like plasmid was carrying the *mphA* and *ermB* gene in a pathogenic gene cluster. The contig carrying other two MRGs (*msrE* and *mphE*) was subjected to NCBI Blast and found that it shows 100% identity to a part of large plasmid

(NZ_CP068889.1) which was also a IncFII type plasmid. Apology for using the confusing terminology “belong to”. In this study we conclude that IncFII plasmid can possess all four MRGs in *Shigella* and possibly the most potential carrier in disseminating the MRGs in *Shigella*. We revised the sentence in the current version (page: 2; line: 38-39).

Comment 2. *Table S2. The total number of isolates is 149. The number of Shigella dysenteriae 2 should be 2. And, the values in the column of "Percentage" should have the same decimal numbers.*

Response: Thanks for your keen observation. We apologize for the error. We corrected the number *Shigella dysenteriae 2* and its frequency (%) in the Supplementary table 1.

Comment 3. *Line 100, Among the 150 Shigella "strains"... "isolate(s) rather than "strain(s)" should be used in most circumstances.*

Response: Thank you for your valuable suggestion. In this current version of this manuscript, we replaced the term “strain(s)” with “isolates in all suitable instances.”

Comment 4. *Line 102, "Supplementary table S2" appears first in the text, it should be revised as table S1.*

Response: Thank you for the keen review. We apologize for the unintentional error. We synchronized the citations and table numbers in the revised manuscript (Page:6; line: 104).

Comment 5. *Table S1. Suggest adding the sizes of amplicons in the Table.*

Response: Thanks for your suggestion. We added the amplicon sizes in the supplementary table 1 and merged it in “Supplementary tables” file as “Supplementary table S4”.

Comment 6. *Line 121, "Azithromycin" Should be replaced by "azithromycin"*

Response: In response to your suggestion, the term “Azithromycin” was replaced with “azithromycin” or “AZM” throughout the manuscript.

Comment 7. *All the supplementary tables (S1, S2, S3) and Supplementary file 1 can be put in an excel file in different spreadsheets.*

Response: Thanks for your valuable suggestion. We put all the supplementary tables in an excel file according to your recommendation (Supplementary tables).

Comment 8. *Line 125, ...(Supplementary file S1). In the excel file, it is marked as "Supplementary file S2".*

Response: We corrected the file name from “Supplementary file S2” to “Supplementary table S2” in the merged file named “Supplementary tables” in the revised version of manuscript.

Comment 9. *Lines 126-127, "PCR of plasmid DNA and genomic DNA for mphA, ermB and msrE genes exhibited the same results" It is confusing! How did the authors separate plasmid DNA from genomic DNA?*

Response: We have extracted genomic DNA and plasmid DNA separately using Wizard® Genomic DNA Purification Kit (Promega) and the PureYield™ Plasmid Miniprep kit System (Promega),

respectively. We performed the PCR for MRGs using DNAs extracted from both genomic and plasmid DNA. We have also detailed description in methodology section (Page: 14; line 283-288).

Comment 10. Lines 134-135, "Moreover, the *mphA* and *ermB* genes were concurrently found in all RSA *Shigella* strains" All the RSA isolates selected for WGS are PCR positive for *mphA* and *ermB*, aren't they?

Response: The line of query specifically describes the 13 isolates we selected for whole genome sequencing based on the presence of both *mphA* and *ermB* genes in PCR study. We paraphrased the line in the revised manuscript (page:7; line: 135-137). Thank you.

Comment 11. Lines 149-157 "Genome BLAST Distance Phylogeny (GBDP)" Is this study relevant to azithromycin resistance?

Response: The samples were collected from same region and within a short period of time (2016-2018), so, there is a high possibility of genetic dissemination of MRGs among different species of *Shigella*. Therefore, we performed the phylogeny analysis to find the clonality among the isolates. For better understanding and to make it more informative, we modified the **Figure 1** in the revised version.

Comment 12. Lines 175-177, "The putative plasmid sequences extracted from *S. flexneri* Z12966 and *S. flexneri* Z13032 were named as pZ12966_11 and pZ13032 respectively (Figure 2)" Without long sequence reads, how did the authors fill the gaps of fragments to obtain complete plasmid sequences? How did the authors make sure the fragments belonged to plasmids instead of chromosomes?

Response: We determined whole plasmid sequences with multiple contigs using specialized software tool "PlasmidSPAdes" for extracting and assembling plasmid data from whole genome sequences (Illumina short reads, paired end). PlasmidSPAdes uses the Illumina read coverage of contigs to distinguish between plasmids and chromosomes (Antipov D et al., 2016).

The resultant multi-contig plasmid sequences were subjected to BLAST analysis against the reference plasmid sequences provided by PlasmidSeeker to affirm the presence of the plasmid of our interest. The detailed procedure has been mentioned in the revised version of the manuscript in methodology section (page 17-18; line 354 – 365) as follows-

"Plasmid sequences were determined using multiple in-silico approaches. PlasmidSPAdes (SPAdes v3.13.0) and PlasmidSeeker v1.3 were used to determine the plasmids from the FASTQ reads (Paired end, R1 and R2). PlasmidSPAdes assembly resulted in plasmid sequences in multiple contigs. PlasmidSeeker provided multiple clusters of reference plasmids that supposed present in the query sequence. The assembled plasmid sequences and whole genome sequences were then used as target sequences in the nucleotide blast (<http://blast.ncbi.nlm.nih.gov/Blast.cgi>) study with reference query sequence of pKSR100. Pangenome and blast atlas were constructed using the Gview tool (<https://server.gview.ca/>) where pKSR100 was used as the reference core for the blast atlas module. Plasmid sequences were typed by PlasmidFinder v2.1 (<https://cge.cbs.dtu.dk/services/PlasmidFinder/>) maintaining 90% threshold identity and 60% minimum coverage."

Reference:

Antipov D, Hartwick N, Shen M, Raiko M, Lapidus A, Pevzner PA. plasmidSPAdes: assembling plasmids from whole genome sequencing data. *Bioinformatics*. 2016 Nov 15;32(22):3380-3387. doi: 10.1093/bioinformatics/btw493. Epub 2016 Jul 27. PMID: 27466620.

Comment 13. The genetic maps in Figure 2 and Figure 3 are not consistent. Suggest simplifying the maps.

Response: As per editor's recommendation, we merged the figure 2 and figure 3 and incorporated as single figure (Figure 2).

Comment 14. Lines 191-192, "These sequences were named as and pZ12966_191 respectively" The 4,952 bp segment should be only part of a plasmid, it should not be named as a plasmid ppZ12959_224 (pZ12966_191).

Response: As per reviewer suggestion, we removed the plasmid name; however, we considered this as contig (4,952 bp) with accession numbers (JAFEJL010000224.1 and JAEUXL010000191.1) (page: 9 ;line: 183-186).

Comment 15. Figure 4 is not necessary, it should be removed.

Response: We removed figure 4 in the revised manuscript.

Comment 16. Figure 5 and Figure S1. Are they relevant to azithromycin resistance?

Response: We agree with reviewer suggestion; therefore, we deleted *Figure 5 and Figure S1* from the revised version of the manuscript.

Comment 17. Lines 196-215, this paragraph is not relevant to azithromycin resistance, it (and Table S3) should be excluded.

Response: Thanks for the valuable suggestion. We deleted both supplementary table S3 and line 196-215 in the revised version of the manuscript.

June 27, 2022

Dr. Zahirul Islam
International Centre for Diarrhoeal Disease Research
Laboratory of Gut-Brain Signaling
68. Tajuddin sarani
Dhaka 1212
Bangladesh

Re: Spectrum00741-22R1 (Multiple Mechanisms Confer Resistance to Azithromycin in *Shigella* in Bangladesh: A Comprehensive Whole Genome-Based Approach)

Dear Dr. Zahirul Islam:

Your manuscript has been accepted, and I am forwarding it to the ASM Journals Department for publication. You will be notified when your proofs are ready to be viewed.

Sincerely,

Monica Garcia-Solache
Editor, Microbiology Spectrum
